# Modeling dynamic social vision highlights gaps between deep learning and humans

**Kathy Garcia**[1*], **Emalie McMahon**[1*], **Colin Conwell**[1], **Michael F. Bonner**[1] **& Leyla Isik**[1,2]
[1]Department of Cognitive Science, [2]Department of Biomedical Engineering
Johns Hopkins University
Baltimore, MD 21218, USA
{kgarci18,emaliemcmahon,cconwell2,mfbonner,lisik}@jhu.edu

[*]authors contributed equally

## ABSTRACT

Deep learning models trained on computer vision tasks are widely considered the most successful models of human vision to date. The majority of work that supports this idea evaluates how accurately these models predict behavior and brain responses to static images of objects and scenes. Real-world vision, however, is highly dynamic, and far less work has evaluated deep learning models on human responses to moving stimuli, especially those that involve more complicated, higher-order phenomena like social interactions. Here, we extend a dataset of natural videos depicting complex multi-agent interactions by collecting human-annotated sentence captions for each video, and we benchmark 350+ image, video, and language models on behavior and neural responses to the videos. As in prior work, we find that many vision models reach the noise ceiling in predicting visual scene features and responses along the ventral visual stream (often considered the primary neural substrate of object and scene recognition). In contrast, vision models poorly predict human action and social interaction ratings and neural responses in the lateral stream (a neural pathway theorized to specialize in dynamic, social vision), though video models show a striking advantage in predicting mid-level lateral stream regions. Language models (given human sentence captions of the videos) predict action and social ratings better than image and video models, but perform poorly at predicting neural responses in the lateral stream. Together, these results identify a major gap in AI's ability to match human social vision and provide insights to guide future model development for dynamic, natural contexts.

## 1 INTRODUCTION

Over the past decade, significant advances have been made in understanding the computations underlying both biological and artificial vision, in large part due to deep learning models that now provide the best match to human visual behavior and neural responses. However, most research has focused exclusively on static scene and object recognition, neglecting the rich, dynamic interactions that characterize real-world vision. Human vision is tuned to process dynamic social scenes from only a few months of age (Hamlin et al., 2007), yet social vision remains a substantial open challenge in artificial intelligence (AI) (Bolotta & Dumas, 2022), where current AI models struggle to match even human infants in their ability to understand social scenes (Gandhi et al., 2022; Shu et al., 2021). Many have argued that incorporating insights from cognitive (neuro)science may improve AI models' performance on social tasks (Zhou et al., 2019; McMahon & Isik, 2023; Malik & Isik, 2023). However, AI vision and language models are rapidly evolving, and current models have never been comprehensively tested against humans in dynamic, social vision.

The human brain processes dynamic social scenes in regions that are distinct from those involved in classical object perception (Tarhan & Konkle, 2020; Wurm et al., 2017; McMahon & Isik, 2023; Lee Masson & Isik, 2021). These regions form the recently proposed lateral visual stream (Pitcher & Ungerleider, 2021; Wurm & Caramazza, 2022), specialized for dynamic social perception and distinct from the classical ventral "what" and dorsal "where" streams (Ungerleider & Mishkin, 1982;

Goodale & Milner, 1992). While AI models effectively predict the ventral stream's computations, little is known about the lateral stream's computations. Recent work suggests computational similarities between the ventral and lateral streams, proposing that lateral stream representations may also be hierarchical, with each computational stage yielding increasingly abstract representations (McMahon et al., 2023). Some work has even suggested that lateral stream computations are not distinct from those in the ventral stream (Finzi et al., 2022; 2023). However, this claim is based on visually-evoked responses to static images, while lateral stream regions respond primarily to dynamic stimuli (Pitcher et al., 2011; Pitcher & Ungerleider, 2021).

Here, we use a large-scale benchmarking approach to investigate the computational principles underlying human social vision and identify areas of critical need for AI model development (Figure 1). Using over 350 image, video, and language models, we predict human behavioral ratings and fMRI responses to a dataset of publicly available natural videos depicting human social actions (McMahon et al., 2023). We find that language models based on sentence captions of the videos are best at predicting human social ratings and that video models are best, on average, at predicting brain responses in the lateral visual stream. However, the performance in lateral stream regions is substantially lower than in the ventral visual stream, and no model is able to match both human behavior and brain data. Together, these results highlight a critical need for image-computable models of social perception that match human abilities.

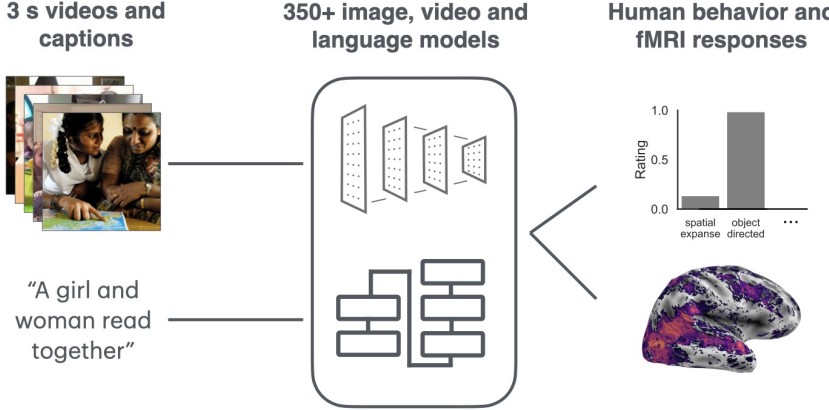

Figure 1: A summary of our overall approach. We extract representations from over 350 image, video, and language models based on three-seconds videos of human social actions or their captions. We then use DNNs to predict human behavioral ratings and fMRI neural responses to the videos.

## 2 RELATED WORK

Our approach builds on the NeuroAI benchmarking approach that has been popularized by others (Schrimpf et al., 2018; 2020; Gifford et al., 2023; Willeke et al., 2022; Conwell et al., 2024; Elmoznino & Bonner, 2024; Chen & Bonner, 2023). For cognitive neuroscience, NeuroAI benchmarking can elucidate the computational factors needed to match the human brain. For AI, these benchmarks can reveal whether different algorithms are human-aligned and suggest avenues for future model development. The aim of NeuroAI benchmarking is often either to identify the single best model of the brain or behavior (Schrimpf et al., 2018; 2020; Gifford et al., 2023; Willeke et al., 2022) or to understand the computational principles underlying human-model alignment (Conwell et al., 2024; Elmoznino & Bonner, 2024; Chen & Bonner, 2023).

Here, we take the latter approach, but rather than aiming to understand static scene responses in the ventral visual stream, which are well modeled by most current image models (Conwell et al., 2024), we aim to understand dynamic visual responses across the brain, focusing on the lateral visual stream and human annotations of the features of interest for these regions. In addition to benchmarking vision models, we also use language models to predict behavioral ratings and visually-evoked neural responses as has been done in prior work with static scenes (Conwell et al., 2023; Doerig et al., 2022;

Geirhos et al., 2021; Linsley et al., 2023; Fel et al., 2022; Lahner et al., 2023). As in Conwell et al. (2023), we use multiple language models as predictors and selectively perturb the sentence captions of the stimuli to provide insights into the kind of linguistic features that are predictive of visual responses. While modeling dynamic visual events is a growing area of interest (Lahner et al., 2023), this is the first investigation of benchmarking many models in response to naturalistic videos of human actions.

## 3 METHODS

### 3.1 CODE AND DATA AVAILABILITY

All code used in this paper and our sentence captions are publicly available: `https://github.com/Isik-lab/SIfMRI_modeling.git`. The social action ratings and fMRI responses are publicly available on OSF `https://osf.io/4j29y/` with a Creative Commons Attribution 4.0 International (CC-BY-4.0) license. The videos shown to participants and used here to extract model activations are from the Moments in Time (MiT) dataset `http://moments.csail.mit.edu`. The MiT license restricts public release of videos from the dataset, but instructions for how to obtain the videos are available in the original manuscript (McMahon et al., 2023).

### 3.2 SOCIAL ACTION DATASET

Here, we model behavioral ratings and neural responses from a publicly available dataset of human social actions (McMahon et al., 2023). All experimental details are included in the original publication. Briefly, participants in the fMRI saw the stimuli at approximately 20 degrees of visual angle and were allowed to freely view the videos. Behavioral ratings were collected online from separate participants.

The dataset includes 250 three-second videos of social actions that are divided into 200 videos for training and 50 videos for evaluation. While this dataset is too small for training most AI models, it is on par with other NeuroAI benchmarking studies Conwell et al. (2024). Each video includes human behavioral ratings of the visual and social scene features. The rated dimensions include descriptions of how large the scene is (*spatial expanse*), how close people in the video are (*interagent distance*), the extent to which the people are facing one another (*agents facing*), the extent to which an action is object-directed (*object directed*), the extent to which people are jointly engaged in an action (*acting jointly*), the extent to which they are communicating (*communicating*), and affective features (*valence* and *arousal*). Ratings were collected on a Likert scale by at least ten subjects, and we use the average rating for each feature. Inter-rater agreement is quite high and is used as the noise ceiling against which our models are evaluated.

The dataset also includes voxel-wise fMRI neural responses to each of the videos (beta values) in four participants, and an estimate of the explainable variance determined as the test-retest reliability of responses to the same videos. To restrict our analyses to reliable voxels, we use the reliability mask for each subject's data provided with the original paper. The fMRI data also includes anatomically (early visual cortex, EVC, and motion-selective middle temporal area, MT) and functionally defined regions of interest (ROIs) in the ventral and lateral streams. The ventral functional ROIs include the face-selective fusiform face area (FFA) (Kanwisher et al., 1997) and place-selective parahippocampal place area (PPA) (Epstein & Kanwisher, 1998), and the lateral ROIs include the extrastriate body area (EBA), which processes bodies and relations between bodies (Downing et al., 2001; Abassi & Papeo, 2020), the lateral occipital cortex (LOC) which is object selective (Grill-Spector et al., 2001) and is involved in processing object-directed actions (Wurm et al., 2017), and posterior and anterior social-interaction selective regions in the STS (pSTS and aSTS) (Isik et al., 2017; Walbrin et al., 2018; Lee Masson & Isik, 2021; McMahon et al., 2023).

### 3.3 SENTENCE CAPTIONING OF VIDEOS

In order to evaluate language model prediction of behavioral and neural responses, we collected sentence captions of the videos from 150 online participants using the Prolific platform and in accordance with the Johns Hopkins Homewood Institutional Review Board. Eligibility criteria included having completed at least 50 tasks with an 85% approval rate, having normal or corrected-to-normal

vision, and being native English speakers. All participants were 18+ (M = 39.72 years old, SD = 13.24) and reported gender and race/ethnicity were as follows: 63 female, 87 male; 114 white, 14 black, 10 asian, 9 mixed race, 2 other, 3 declined to report. Participants were compensated approximately $12 per hour on average for their participation.

Following informed consent, each participant captioned 12 videos presented in a random order: 10 videos from the dataset used in McMahon et al. (2023) and 2 additional catch videos that were the same across all participants. Each video appeared with a text box with grayed-out text that read "Description of the actions and interactions of the people in the video in a single sentence..." and disappeared when the subject began typing their caption. Because the study was conducted online and we did not restrict the device or browser used by participants, the visual presentation likely varied across participants.

We collected at least five unique captions for each video in the main dataset. Captions were cleaned by calculating the Pearson correlation distance between each participant and all other participants for the catch video captions in the embedding space of Hugging Face's fine-tuned all-MiniLM-L12-v1 (Wolf et al., 2020; Wang et al., 2020). If a participant's mean distance was outside of 2.5 standard deviations of the mean distance of all other participants for either of the two catch videos, they were removed from all subsequent analyses. All-MiniLM-L12-v1 was not reused in subsequent analyses.

## 3.4 DNN MODEL SELECTION

We used an "opportunistic" modeling approach from other recent NeuroAI benchmarking research Conwell et al. (2024). We selected a large set of publicly available models with a variety of modalities, architectures, training sets and objectives, and evaluated their performance along each of these dimensions. Image and video models were selected to represent a comprehensive cross-section of high-level visual tasks, including category supervision, self-supervision, and multimodal (image-language) training, and also include convolutional and transformer architectures. In total, we tested 348 image models from collections including Torchvision (TorchVision, 2016) and Pytorch-Image-Models libraries (Wightman, 2019), VISSL (Goyal et al., 2021), OpenAI CLIP (Radford et al., 2021), and Dectectron2 (Wu et al., 2019), and 8 video models, including Facebook's SlowFast (Feichtenhofer et al., 2018) and TimeSformer (Bertasius et al., 2021) models. Language models included 20 sentence-transformers and embeddings-based models, such as GPT-2 (Radford et al., 2019), BERT variants (Devlin et al., 2019). To additionally test the effect of large language models, we used the API from Cohere and OpenAI to extract embeddings for Cohere's English v3 classification and clustering models, and GPT-3 large embeddings, GPT-3 small embeddings, and GPT Ada-002. For a full model list and the corresponding license information, please see the supplemental files. The vision and language embeddings for multimodal models were considered separately in model evaluation (e.g., CLIP image encoder is grouped with image models and CLIP's text encoder with language models). We tested fewer video and language models relative to image models due to their availability and computational requirements, respectively. However, this only strengthens our conclusions when either model class outperforms image models.

## 3.5 MODEL ALIGNMENT WITH BEHAVIORAL AND NEURAL RESPONSES

### 3.5.1 MODEL FEATURE EXTRACTION

We utilized DeepJuice (Conwell et al., 2024), a python package in alpha-release shared with us by the authors, that allows for memory-efficient feature extraction from each layer of a DNN. We extracted the intermediate representations from every unique computational submodule (referred to here as layers) of every model, with the exception of the non-open models from Cohere and OpenAI where we could only extract a single embedding via the API. We then used GPU-optimized sparse random projection (SRP) implemented in the python package to project the activations in an approximately 4732-dimensional feature space based on the Johnson–Lindenstrauss lemma with $\epsilon = 0.1$ (Larsen & Nelson, 2014).

All model inputs were preprocessed in a model-specific manner. For image models, we extracted activations for seven evenly-sampled frames across the three seconds of each video and then averaged the activations across frames from the same video. In preliminary analyses, we found that this produced almost identical results to using activations from only a single frame or concatenating

activations across the seven frames. Similar to image models, for language models, we extracted the activations for each caption, and then averaged the activations for the captions from the same video.

### 3.5.2 Linear mapping

Before fitting the linear mapping, we first Z-scored the model-SRP feature space across the samples independently for each feature in the 200-video train set defined in the original dataset (McMahon et al., 2023) and then normalized the held-out data from 50 videos by the mean and standard deviation from the train set. We normalized the behavioral and neural data using the same procedure.

We performed linear mapping between the normalized model-SRP feature space and the normalized behavioral or neural response using leave-one-out ridge regression optimized for the GPU as implemented in DeepJuice (Conwell et al., 2024). Our $\alpha$-penalty search space was seven values sampled from a logspace of $10e^{-2}$ to $10e^5$. In the training set, we performed 4-fold cross-validation in a full sweep of the model to determine the layer that produced the highest performance on the held-out data. Performance was measured as the Pearson correlation between the predicted behavioral or neural response and the true response.

We selected the optimal model layer based on this cross-validation in the training set, and evaluated each model's performance for the optimal layer in the test set. The optimal model layer was selected separately for every behavioral rating and voxel in the brain. For the brain data, we only predicted responses in voxels that were determined to have high test-retest reliability in the original dataset.

### 3.6 Caption perturbation experiment

To gain insight into what aspects of the captions produced high alignment between the language models and the brain/behavior, we performed selective perturbations to the captions as in (Conwell et al., 2023; Kauf et al., 2024), but instead of deletion, we performed masking to keep the overall syntactic structure of the sentence intact. We used the spaCy (Honnibal & Montani, 2017) package and the en_core_web_sm model in particular to identify the parts of speech for masking. All other aspects of the experiment followed our model-brain/behavior mapping procedure described above.

### 3.7 Statistical analysis

To determine whether there was a difference in mean performance between classes of models (e.g., image relative to language), we first computed a null distribution for each model by correlating the permuted predicted response and actual response over 5,000 iterations. The same shuffling procedure was used across all models. We compared the true mean difference between model classes to a null distribution of mean differences. The *p*-value was determined by performing a two-tailed test of the true value against the permuted distributions.

The same procedure was used to determine whether a perturbation on the sentence captions significantly decreased alignment between the language models and the brain/behavior, except using a one-tailed test to compare true degradation (performance on the "original" minus performance on the perturbed sentence) to chance.

Models that were not publicly available (e.g., Cohere variants) were not included in the statistical testing because we were not able to do a full sweep of these models. Note that these models do not strongly outperform other models in the brain 2 or behavior 1, so this is unlikely to affect any conclusions from the paper.

### 3.8 Compute specifications

The ridge regression for the encoding models required a substantial amount of computational resources. We used an institutional high-performance computing cluster equipped with 31 A100 GPU nodes (with a mix of 40 and 80 GB memory). On average, each set of model regressions took approximately 0.38 core hours for the whole-brain results and 0.39 core hours for behavior rating results, for a total of 0.77 core hours per model. To run the full suite of 365 image, video, and language models for the reported results took approximately 280 core hours. Full computational

resources for the research project (including failed experiments and experiments not reported here) required approximately 1600 core hours.

## 4 RESULTS

### 4.1 BEHAVIORAL RATINGS

#### 4.1.1 LANGUAGE MODELS CAPTURE HUMAN VISUAL SOCIAL RATINGS BETTER THAN VISION MODELS

In evaluating the models, we can compare both how the different model classes (image, video, or language) perform on average at predicting each caption (Figure 2) and the top performing model for each rating (Supplemental Table 1). We note that due to the larger number of image models tested, the best performing model is biased towards the image models. Despite this, we see a large amount of similarity between the two metrics.

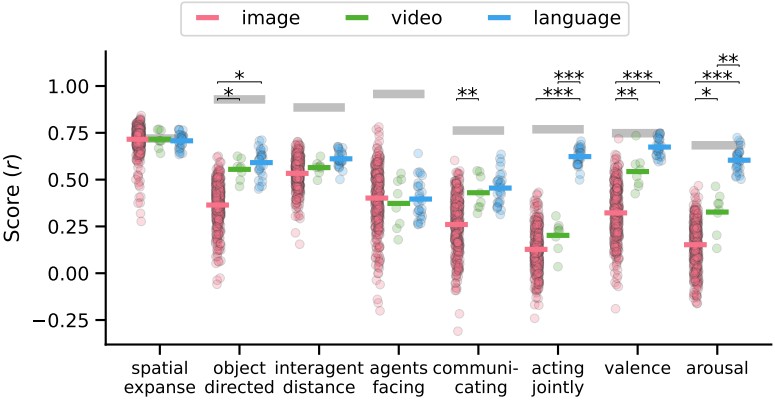

Figure 2: Prediction performance of all models in predicting behavioral responses. Each dot is the performance of a single model. The colored horizontal lines indicate the mean performance for image (pink), video (green), and language (blue) models. The horizontal gray lines are the inter-subject agreement, which is approximately the maximal level that any model could be expected to perform. Brackets and asterisks indicate significantly different performance between different classes of models ($p < 0.05$: *, $p < 0.01$: **, $p < 0.001$: ***).

We find that for the visuospatial ratings (spatial expanse, interagent distance, and agents facing), no model class is substantially better on average ($p > 0.05$), but for each rating, the top performing model is an image model. In contrast, for all social and action ratings, the best performing model is a language model despite the over-representation of image models in our model set. Though we note the larger, more modern models did not perform best for most ratings (Supplemental Figure 12, Table 1). For predicting ratings of object directed actions and communicating, the mean difference between language and vision models is not significant ($p > 0.05$). For ratings of agents acting jointly and affective features (valence and arousal), language models perform better on average than image models ($ps < 0.01$) and video models ($ps < 0.001$), except for valence where language models do not perform better than video models ($p > 0.05$).

For most ratings, video models do not perform better than image models, except for predictions of communicating, valence, and arousal ($ps < 0.05$). For image models, we do not see different predictions based on a convolutional versus transformer architecture (Supplemental Figure 8), supervised versus self-supervised learning objective (Supplemental Figure 7), language-aligned training (Supplemental Figure 6), training dataset (Supplemental Figure 9), training objective (Supplemental Figure 11), or the number of tunable parameters in a model (Supplemental Figure 10). Together, these results suggest that many drastically different vision models perform similarly in predicting behavioral ratings.

### 4.1.2 HUMAN-LANGUAGE MODEL ALIGNMENT DEPENDS ON BOTH NOUN AND VERB CONTENT

To understand the features driving the relatively high performance of language models, we performed selective perturbations on sentence captions by removing nouns or verbs or leaving only nouns and verbs from the captions (Figure 3A). We calculate the degradation in performance as the score of the original, unperturbed input captions ($r_o$) minus the score of the perturbed input captions ($r_p$) divided by the score of the original ($r_o$).

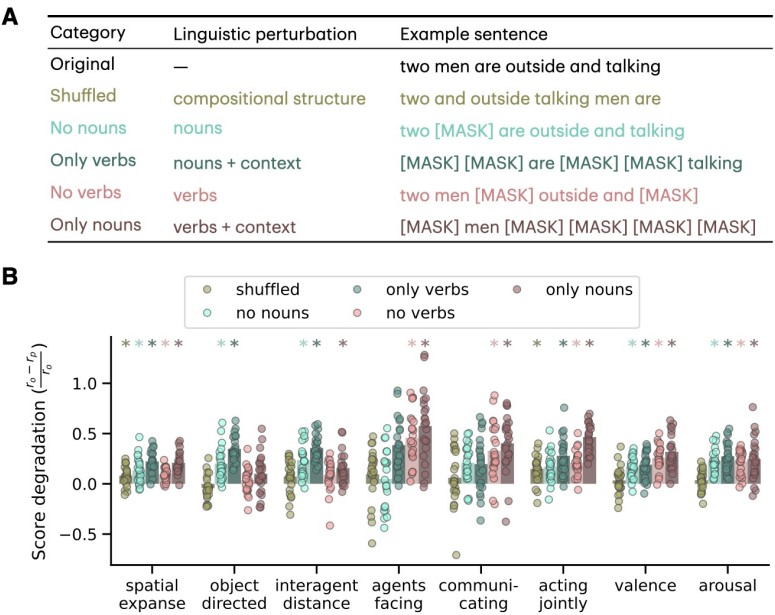

Figure 3: **A**. Example sentence perturbations. **B**. The performance of each language model (dots) in predicting human behavioral ratings following selective perturbation of the sentence captions. The bars indicate the mean performance across models for each condition and rating. Asterisks indicate that there is a significant degradation in model-behavioral alignment following perturbation relative to the unperturbed sentence.

We find that when predicting most ratings shuffling the input captions does not decrease performance, except spatial expanse and agents acting jointly ($p < 0.05$), suggesting that behavior-model alignment does not rely on linguistic compositional structure (Figure 3B).

We can group the remaining perturbations based on disruption to noun content (no nouns, only verbs, teals in Figure 3B) and disruptions to verb content (no verbs, only nouns, pinks in Figure 3B). Prediction of most ratings is degraded by disrupting noun content ($ps < 0.05$), except for agents facing and communicating. Prediction of most ratings is also degraded by both verb manipulations (no verbs and only nouns, $ps < 0.05$), except ratings of object directed actions ($p > 0.05$) and interagent distance ($ps > 0.05$). Together, these results suggest that the success of language models in predicting visual responses relies on both noun and verb content, with the exception of agents facing/communication and object-directedness, respectively.

## 4.2 NEURAL RESPONSES

### 4.2.1 VISION MODELS BEST CAPTURE NEURAL RESPONSES

As in behavior, we can compare both the average performance of the models, and the best model for each ROI (Supplemental Table 2). We evaluate performance in ROIs (Figure 4) and the whole brain (Figure 5). We find that for several mid-level ROIs (MT, EBA, and LOC) and a high-level ROI (pSTS), video models outperform image models on average ($ps < 0.001$). In both early visual cortex and aSTS, the quantitative performance gain is moderate and not significantly different ($ps$

> 0.05), and the best performing model is an image model. Within image models, we we did not find a notable difference in performance based on architecture (Supplemental Figure 16), training task (Supplemental Figures 17, 15, 21), or number of tunable parameters (Supplemental Figure 20). While there is an improvement of kinetics-trained models over other models in mid-level regions (Supplemental Figure 19), kinetics-trained models are all video models and the other datasets are all used for training image models. Language models, including large language models (GPT-3 and Cohere, Supplemental Figure 22), performed relatively poorly overall. Together, these results further underscore the importance of video processing as a critical computational factor within vision models for human alignment.

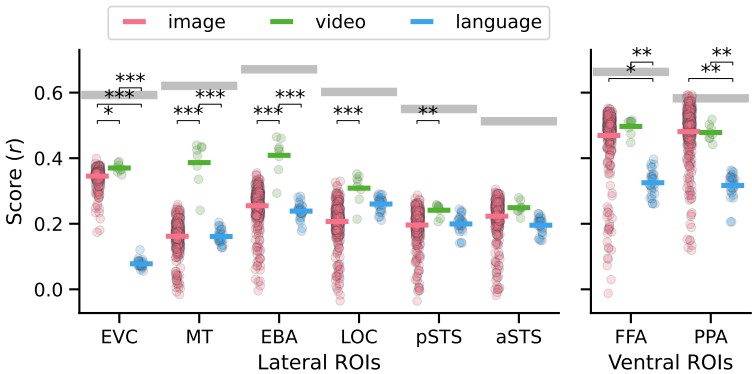

Figure 4: The performance of each model (dots) in predicting the average response in each ROI. The colored lines indicate the mean performance of the different classes of models, and the horizontal gray line is the split-half reliability of the voxel responses in each ROI averaged across participants. Brackets and asterisks indicate significantly different performance between different classes of models ($p < 0.05$: *, $p < 0.01$: **, $p < 0.001$: ***).

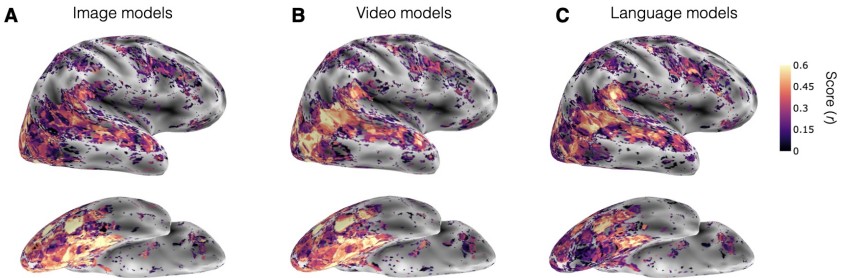

Figure 5: Visualization of the test set encoding performance of the best performing layer in the training set for each voxel from any (**A**) image, (**B**) video, and (**C**) language model. This is shown on the lateral and ventral surface in right hemisphere of one representative participant.

Contrary to what we see in behavior, language models do not outperform vision models on average in any region ($ps > 0.05$), but they significantly underperform in EVC, FFA, and PPA ($ps < 0.05$), and there is no ROI in which the best model is a language model. Language perturbation experiments suggest both noun and verb content are important for prediction in most ROIs (Figure 18).

Despite the relatively higher performance by video models in predicting mid-level lateral regions, we still see a striking under performance of even the best models in lateral regions relative to ventral regions (Figures 4, 5).

### 4.2.2 HIERARCHICAL ALIGNMENT BETWEEN MODELS AND BRAINS

Previous investigations have found a hierarchical correspondence between vision models and the brain in the ventral temporal cortex in humans (Khaligh-Razavi & Kriegeskorte, 2014) and inferiortemporal cortex in macaques (Yamins et al., 2014). In these studies, early model layers provide the best match to posterior brain regions and later layers better match more anterior regions. We investigated whether there was a similar hierarchical correspondence between models and responses in lateral visual regions by evaluating the relative depth of each voxel in the whole brain (Supplemental Figure 13) and on average in our regions of interest (Supplemental Figure 14). We find that though early visual cortex is best predicted by earlier layers in the models, all other regions are predicted by layers of approximately equal depth. In contrast, whole brain results do show a hierarchy along the ventral stream, thus highlighting an additional gap between models and lateral stream regions.

## 5 DISCUSSION

We used a large set of image, video, and language models to predict human behavioral ratings and and neural responses to dynamic social scenes. Overall, we found a notable gap in all models' ability to predict human responses. However, there were differences in the models that are best able to predict the brain versus behavior. In particular, language models tended to be the best models of human behavioral ratings, while video models best predicted responses in lateral brain regions.

### 5.1 LANGUAGE AND VIDEO MODELS FOR SOCIAL SCENE UNDERSTANDING

The fact that language models align with human social ratings may suggest that humans rely on non-visual aspects of social interactions to rate social features (McMahon et al., 2023; Netanyahu et al., 2021). While this may be partially true, it is unlikely to be the whole answer because humans make many of these social judgments quickly and automatically (McMahon & Isik, 2023; Quadflieg & Penton-Voak, 2017). Further, these behavioral ratings strongly predict visual regions of the brain (McMahon et al., 2023; McMahon & Isik, 2023), which the language models cannot explain. Another reason the language models predict behavior so well may be due to the captioning prompt. By instructing participants to caption the "actions and interactions" in the videos, we may have biased the language model embeddings to better predict social action content. It is therefore somewhat surprising that language models do not perform even higher than reported here because, for example, high-communication videos are often captioned explicitly with verbs like "talking" (e.g., example caption in Figure 3), and yet most language models still fall short of human agreement.

In contrast, video models provided a boost to prediction in mid-level lateral regions, which is surprising given the lack of high-performing video models relative to image models in our set, but they did not predict behavioral ratings or more anterior regions significantly better than image models. Together, this work reveals a significant gap in even state-of-the-art models' abilities to match human social action judgments and the underlying neural substrates. These results also provide insights into future directions for model development that can integrate the relational structure that is readily present in language with dynamic visual information.

### 5.2 NEUROAI IN DYNAMIC SOCIAL CONTEXTS

One major advantage of this work over most prior NeuroAI studies is the focus on dynamic, social scenes. In addition to the human-model gap, several other interesting findings come out of testing models in more ecologically valid conditions. First, prior work has suggested that affective features like valence and arousal can be extracted by relatively simple convolutional neural networks (Kragel et al., 2019; Conwell et al., 2021). While these features may be image computable, our work shows that in dynamic events, few existing vision models can match human social-affective ratings.

Further, unlike prior work with static scenes (Conwell et al., 2023; Doerig et al., 2022), language models were not able to capture brain responses in the current dataset. They perform dramatically worse than image models in predicting responses in EVC, as has been documented elsewhere (Conwell et al., 2023), but also significantly worse in high-level ventral regions (Figures 4, 5). This result calls into question strong notions of "language alignment" in visual cortex and highlights the importance of dynamic stimuli for studies of even the ventral visual stream (Haxby et al., 2020).

### 5.3 LIMITATIONS

One limitation of the current study is that the relatively small size of our dataset means it is not suitable for training AI models. Larger, more diverse datasets that can better represent the complexity of real-world social interactions are still needed. While some dynamic NeuroAI datasets have started to be released (Lahner et al., 2023; Zhou et al., 2023), they do not rival the stimulus variety and dataset quality of static image NeuroAI datasets (Allen et al., 2022). Similarly, computer vision video datasets (Kay et al., 2017; Monfort et al., 2019; Thomee et al., 2016) also lack the size and diversity of image datasets (Deng et al., 2009; Thomee et al., 2016).

Another limitation is that this work—as do most NeuroAI benchmarking studies—takes advantage of available models that differ along many factors, making it difficult to isolate the impact of any one computational factor on performance. This limitation may explain why we do not see significant differences for seemingly important model factors, such as convolutional versus transformer architectures (Supplemental Figures 8, 16). We aimed to overcome this by testing a wide array of models, but as noted above, our set includes relatively few video and language models compared to image models. Future work should investigate how larger language models compare to human on social visual tasks, though we note here that the larger language models tested (GPT and Cohere variants) were far from the top performing model in most cases (Supplemental Tables 1, 2) and models with more tunable parameters did not always yield higher human data predictivity (Supplemental Figures 10, 20).

### 5.4 FUTURE DIRECTIONS

Moving forward, we suggest a couple avenues for advancing model development based on these results. The relative success of language and video models in matching behavior and brain respectively, suggest that models that explicitly represent agents, objects, and their relations over time will be critical to future modeling endeavors. While transformer models should be able to capture some of this relational information in theory, they still fall short in predicting the brain and behavior in the current dataset. This failure may be due to the datasets and tasks they are trained on, which do not require the agent and object-based representations that humans deploy in social reasoning tasks. It is possible that training on more human-aligned tasks will allow these large transformer models to pick up on relevant structure in social scenes.

Human-aligned DNNs may be a promising direction for dynamic social perception. For example, prior work has shown that implementing neuroscience-inspired circuits into transformers provides more human-aligned object tracking (Linsley et al., 2021). Other work has focused on training modern DNNs with more ecologically valid data. For example, Mineault et al. (2021) trained a 3D ResNet model on agent self-motion to model dorsal stream responses, while Orhan & Lake (2024) used infant headcam data to train models on high-level visual representations. We included the latter models in our set, and while they performed relatively well in our benchmarks, they were still not able to maximally predict human social ratings or neural responses (Supplemental Tables 1, 2).

Finally, efforts to merge structured cognitive modeling (Malik & Isik, 2023; Shu et al., 2021; Netanyahu et al., 2021) with image-computable DNNs may be another fruitful direction. One example model, a combined GNN-RNN, tracks entities and their relations over time in a similar video dataset to predict human social judgments (Malik & Isik, 2023). Related prior work has also suggested that generative inverse planning models may be helpful in closing the gap between human social perception and current DNNs (Shu et al., 2021; Netanyahu et al., 2021). While most instantiations of these models are not image computable, this is an active area of model development that we believe is likely to yield advances in matching human social perception (Shi et al., 2024).

### 5.5 CONCLUSIONS

This work identifies human social vision as a key area of need for future AI research. It also offers some promising directions for future modeling endeavors. In particular, the relative success of language and video models over image models in predicting social behavior and brain responses suggests that models combining both compositional and dynamic information may be critical for more human-like social AI.

REPRODUCIBILITY STATEMENT

To ensure the reproducibility of our results we have made our code base publicly available on GitHub at [`https://github.com/Isik-lab/SIfMRI_modeling`], which includes the code and information on pre-processing steps, feature extraction, linear mapping, statistical analyses and model zoo and licenses for all models tested. The experiments are reproducible using the steps provided in the README file. The code is dependent on one additional Python library DeepJuice that is available upon request. Instructions to obtain this library are also included in the README file.

ETHICS STATEMENT

The research adheres to all relevant ethical guidelines. The human subjects experiments conducted for this paper were done in accordance with procedures approved by our institutional IRB. The original publicly available neuroimaging dataset was also conducted in accordance with IRB, and all human subjects were compensated for their time. The model benchmarking used open source models and all model repositories and license details are included with supplementary files.

ACKNOWLEDGMENTS

This work was funded in part by NSF GRFP DGE-2139757 awarded to K.G., NSF GRFP DGE-1746891 awarded to E.M., and NIMH R01MH132826 awarded to L.I.

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

## A  SUPPLEMENTAL MATERIALS

Table 1: A table of the top-30 performing models averaged across features. Models are arranged in descending order of their overall average performance. Bold scores highlight the model that is the top performing model for a given feature, and the absence of a bold score for a given feature indicates that the top performing model is not in the overall top-30 performing models.

| Row number | Model name | Model class | spatial expanse | object directed | interagent distance | agents facing | communicating | acting jointly | valence | arousal |
|---|---|---|---|---|---|---|---|---|---|---|
| 1 | paraphrase-MiniLM-L6-v2 | language | 0.764 | 0.576 | 0.664 | 0.534 | **0.635** | 0.583 | 0.692 | 0.684 |
| 2 | all-mpnet-base-v2 | language | 0.688 | 0.665 | 0.676 | 0.537 | 0.384 | 0.678 | 0.743 | **0.725** |
| 3 | FacebookAI_roberta-large-mnli | language | 0.737 | 0.557 | 0.599 | 0.640 | 0.572 | 0.648 | 0.666 | 0.614 |
| 4 | paraphrase-multilingual-MiniLM-L12-v2 | language | 0.725 | 0.609 | 0.621 | 0.523 | 0.595 | 0.625 | 0.684 | 0.647 |
| 5 | all-mpnet-base-v1 | language | 0.766 | **0.711** | 0.615 | 0.424 | 0.313 | 0.642 | 0.747 | 0.709 |
| 6 | mixedbread-ai_mxbai-embed-2d-large-v1 | language | 0.766 | 0.520 | 0.670 | 0.541 | 0.431 | **0.706** | 0.687 | 0.588 |
| 7 | gpt3_large_embeddings | language | 0.769 | 0.707 | 0.605 | 0.382 | 0.530 | 0.642 | 0.621 | 0.633 |
| 8 | all-distilroberta-v1 | language | 0.719 | 0.680 | 0.603 | 0.461 | 0.494 | 0.607 | 0.629 | 0.611 |
| 9 | paraphrase-multilingual-mpnet-base-v2 | language | 0.717 | 0.571 | 0.632 | 0.470 | 0.499 | 0.579 | 0.715 | 0.564 |
| 10 | gpt_ada_2_embeddings | language | 0.681 | 0.659 | 0.594 | 0.263 | 0.612 | 0.660 | 0.609 | 0.635 |
| 11 | distiluse-base-multilingual-cased-v1 | language | 0.740 | 0.630 | 0.669 | 0.266 | 0.489 | 0.582 | 0.656 | 0.651 |
| 12 | clip-ViT-B-32-multilingual-v1 | language | 0.709 | 0.607 | 0.545 | 0.400 | 0.483 | 0.669 | 0.724 | 0.541 |
| 13 | clip_vitl14 | image | 0.771 | 0.535 | 0.660 | **0.781** | 0.492 | 0.319 | 0.718 | 0.392 |
| 14 | mixedbread-ai_mxbai-colbert-large-v1 | language | 0.660 | 0.540 | 0.570 | 0.432 | 0.528 | 0.670 | 0.652 | 0.599 |
| 15 | gpt3_small_embeddings | language | 0.735 | 0.662 | 0.567 | 0.337 | 0.378 | 0.610 | 0.713 | 0.643 |
| 16 | mixedbread-ai_mxbai-embed-large-v1 | language | 0.690 | 0.558 | 0.633 | 0.465 | 0.461 | 0.630 | 0.631 | 0.570 |
| 17 | gpt2 | language | 0.719 | 0.649 | 0.592 | 0.326 | 0.456 | 0.569 | **0.749** | 0.564 |
| 18 | FacebookAI_roberta-base | language | 0.730 | 0.494 | 0.654 | 0.481 | 0.437 | 0.498 | 0.681 | 0.600 |
| 19 | all-MiniLM-L6-v2 | language | 0.700 | 0.627 | 0.579 | 0.343 | 0.481 | 0.529 | 0.661 | 0.636 |
| 20 | stsb-distilroberta-base-v2 | language | 0.668 | 0.620 | 0.600 | 0.388 | 0.354 | 0.579 | 0.720 | 0.624 |
| 21 | all-MiniLM-L6-v1 | language | 0.634 | 0.605 | 0.676 | 0.260 | 0.492 | 0.611 | 0.596 | 0.637 |
| 22 | multi-qa-MiniLM-L6-cos-v1 | language | 0.686 | 0.461 | 0.648 | 0.337 | 0.347 | 0.643 | 0.668 | 0.700 |
| 23 | cohere_english_v3_classification | language | 0.641 | 0.633 | 0.598 | 0.305 | 0.422 | 0.664 | 0.648 | 0.520 |
| 24 | cohere_english_v3_clustering | language | 0.640 | 0.632 | 0.597 | 0.305 | 0.424 | 0.663 | 0.648 | 0.520 |
| 25 | FacebookAI_xlm-roberta-large | language | 0.734 | 0.614 | 0.600 | 0.328 | 0.355 | 0.609 | 0.601 | 0.570 |
| 26 | all-roberta-large-v1 | language | 0.739 | 0.489 | 0.602 | 0.412 | 0.337 | 0.687 | 0.634 | 0.502 |
| 27 | LaBSE | language | 0.648 | 0.550 | 0.563 | 0.274 | 0.476 | 0.674 | 0.625 | 0.568 |
| 28 | FacebookAI_xlm-roberta-base | language | 0.669 | 0.450 | 0.640 | 0.372 | 0.379 | 0.581 | 0.738 | 0.534 |
| 29 | timm_deit3_large_patch16_224_in21ft1k | image | 0.820 | 0.504 | 0.663 | 0.628 | 0.371 | 0.340 | 0.629 | 0.300 |
| 30 | all-MiniLM-L12-v2 | language | 0.731 | 0.465 | 0.501 | 0.288 | 0.373 | 0.603 | 0.733 | 0.515 |

Table 2: A table of the top-30 models on averaged across all ROIs. Models are arranged in descending order of their overall average performance in all reliable voxels. Bold scores highlight the top score in each ROI. The bottom row indicates the performance in predicting the ROIs based on hand engineered features from (McMahon et al., 2023).

| Rank | Model name | Model class | EVC | MT | EBA | LOC | pSTS | aSTS | FFA | PPA |
|---|---|---|---|---|---|---|---|---|---|---|
| 1 | x3d_s | video | 0.390 | **0.440** | 0.427 | **0.352** | 0.251 | 0.249 | 0.510 | 0.518 |
| 2 | timm_beitv2_base_patch16_224 | image | 0.369 | 0.245 | 0.349 | 0.326 | 0.275 | 0.293 | 0.536 | 0.588 |
| 3 | timm_beitv2_large_patch16_224 | image | 0.376 | 0.259 | 0.355 | 0.330 | **0.287** | 0.291 | 0.524 | 0.593 |
| 4 | x3d_m | video | 0.386 | 0.425 | 0.460 | 0.336 | 0.255 | 0.280 | 0.512 | 0.492 |
| 5 | timm_beit_large_patch16_224 | image | 0.373 | 0.241 | 0.335 | 0.314 | 0.277 | **0.297** | 0.520 | 0.592 |
| 6 | timm_beit_large_patch16_384 | image | 0.368 | 0.251 | 0.336 | 0.317 | 0.253 | 0.266 | 0.501 | **0.593** |
| 7 | clip_vitl14 | image | 0.376 | 0.223 | 0.325 | 0.289 | 0.245 | 0.278 | 0.526 | 0.574 |
| 8 | i3d_r50 | video | 0.378 | 0.433 | **0.465** | 0.345 | 0.254 | 0.239 | 0.516 | 0.442 |
| 9 | timm_deit3_huge_patch14_224_in21ft1k | image | 0.375 | 0.233 | 0.320 | 0.291 | 0.264 | 0.287 | 0.538 | 0.587 |
| 10 | timm_deit3_large_patch16_384_in21ft1k | image | 0.376 | 0.241 | 0.328 | 0.292 | 0.253 | 0.276 | 0.517 | 0.574 |
| 11 | timm_beit_base_patch16_224 | image | 0.365 | 0.228 | 0.322 | 0.298 | 0.258 | 0.267 | 0.522 | 0.578 |
| 12 | timm_deit3_large_patch16_224_in21ft1k | image | 0.371 | 0.241 | 0.337 | 0.295 | 0.245 | 0.274 | 0.499 | 0.584 |
| 13 | slow_r50 | video | 0.358 | 0.408 | 0.421 | 0.329 | 0.248 | 0.245 | 0.512 | 0.477 |
| 14 | timm_convnext_large | image | 0.373 | 0.230 | 0.318 | 0.284 | 0.252 | 0.273 | 0.502 | 0.559 |
| 15 | timm_convnext_large_in22ft1k | image | 0.373 | 0.230 | 0.318 | 0.284 | 0.252 | 0.273 | 0.502 | 0.559 |
| 16 | timm_convnext_xlarge_in22k | image | 0.370 | 0.219 | 0.303 | 0.283 | 0.256 | 0.265 | 0.515 | 0.555 |
| 17 | c2d_r50 | video | 0.371 | 0.376 | 0.405 | 0.305 | 0.258 | 0.272 | 0.511 | 0.462 |
| 18 | timm_deit3_medium_patch16_224_in21ft1k | image | 0.363 | 0.228 | 0.348 | 0.284 | 0.249 | 0.270 | 0.522 | 0.568 |
| 19 | timm_convnext_xlarge_in22ft1k | image | 0.372 | 0.233 | 0.320 | 0.285 | 0.247 | 0.268 | 0.501 | 0.558 |
| 20 | clip_rn50x4 | image | 0.376 | 0.229 | 0.316 | 0.277 | 0.237 | 0.272 | **0.552** | 0.568 |
| 21 | timm_convnext_large_in22k | image | 0.368 | 0.225 | 0.307 | 0.269 | 0.249 | 0.282 | 0.520 | 0.555 |
| 22 | timm_mixer_b16_224_miil_in21k | image | **0.399** | 0.219 | 0.304 | 0.254 | 0.238 | 0.292 | 0.542 | 0.559 |
| 23 | timm_beit_base_patch16_384 | image | 0.358 | 0.230 | 0.331 | 0.292 | 0.235 | 0.266 | 0.512 | 0.568 |
| 24 | timm_convnext_base_in22ft1k | image | 0.364 | 0.225 | 0.324 | 0.275 | 0.240 | 0.258 | 0.517 | 0.549 |
| 25 | timm_convnext_base | image | 0.364 | 0.225 | 0.324 | 0.275 | 0.240 | 0.258 | 0.517 | 0.549 |
| 26 | timm_deit3_base_patch16_224_in21ft1k | image | 0.362 | 0.217 | 0.321 | 0.286 | 0.224 | 0.257 | 0.539 | 0.576 |
| 27 | slowfast_r50 | video | 0.363 | 0.435 | 0.432 | 0.313 | 0.242 | 0.250 | 0.474 | 0.461 |
| 28 | clip_vitb32 | image | 0.376 | 0.203 | 0.300 | 0.246 | 0.262 | 0.241 | 0.507 | 0.568 |
| 29 | timm_deit3_base_patch16_384_in21ft1k | image | 0.367 | 0.209 | 0.289 | 0.269 | 0.248 | 0.269 | 0.526 | 0.566 |
| 30 | timm_convnext_base_in22k | image | 0.362 | 0.217 | 0.308 | 0.276 | 0.228 | 0.250 | 0.515 | 0.559 |
| - | McMahon et al. (2023) | hand engineered | 0.347 | 0.392 | 0.365 | 0.336 | 0.261 | 0.292 | 0.418 | 0.435 |

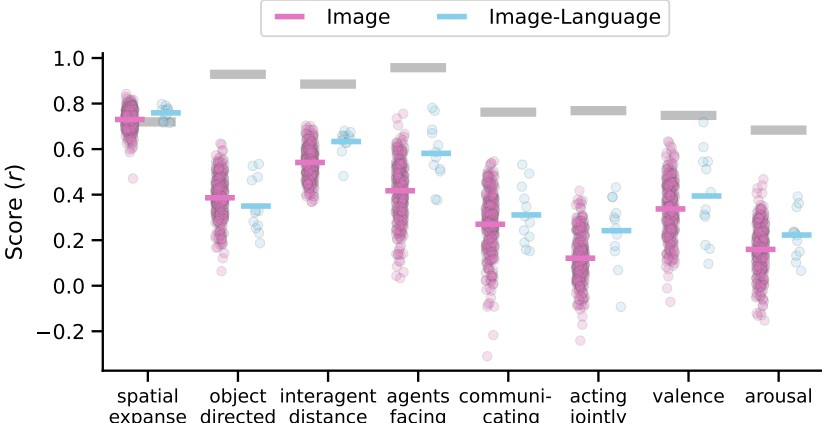

Figure 6: The performance of models in our set trained either with an image-based (image, pink) or multimodal (image-language, blue) objective function in predicting behavior ratings. Plotting conventions are the same as Figure 2.

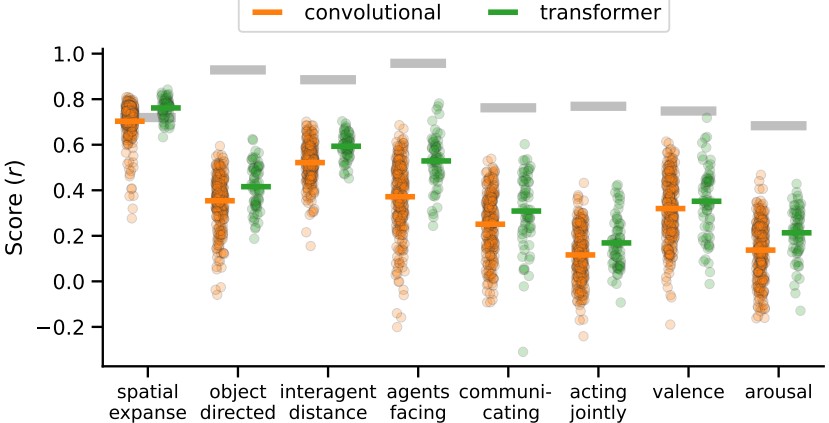

Figure 7: The performance of each image model in our set in predicting behavioral responses separated by whether the model has a convolutional (orange) or transformer (green) architecture. Plotting conventions are the same as Figure 2.

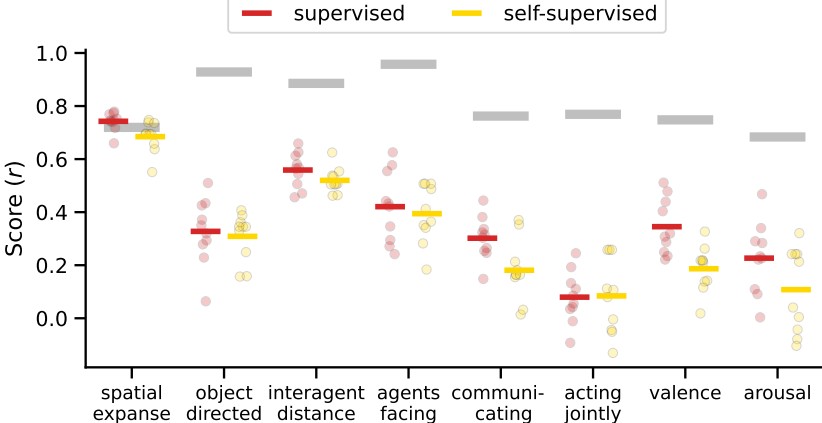

Figure 8: The performance of each image model in our set with a ResNet-50 backbone in predicting behavioral responses separated by whether the model uses a supervised (red) or self-supervised (yellow) learning objective. This analysis was restricted to a single architecture class (ResNet-50) to focus specifically on the learning objective. Plotting conventions are the same as Figure 2.

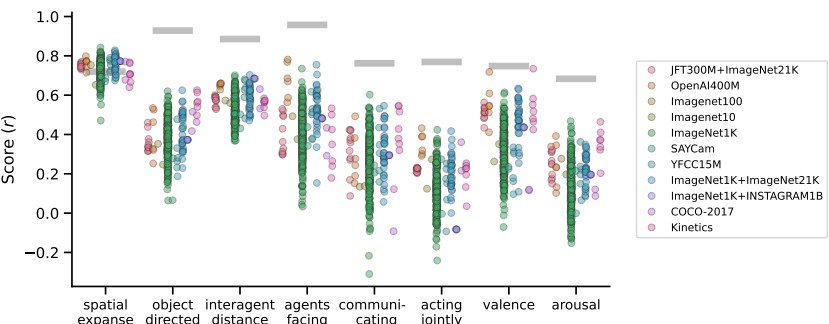

Figure 9: The performance of the vision models (image and video) at predicting the behavioral responses grouped according to their training data. Plotting conventions are the same as Figure 2.

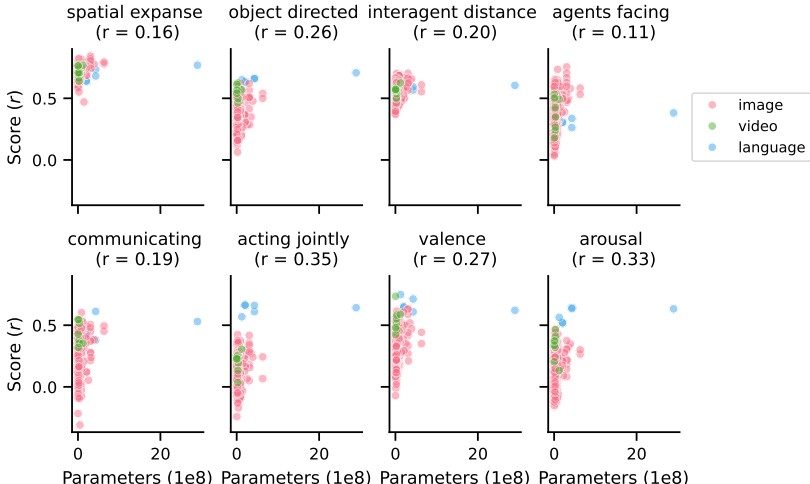

Figure 10: For each behavioral response, the model's prediction score is plotted against the number of trainable parameters. The r-value below the behavioral response indicates the Pearson correlation between the score and number of trainable parameters.

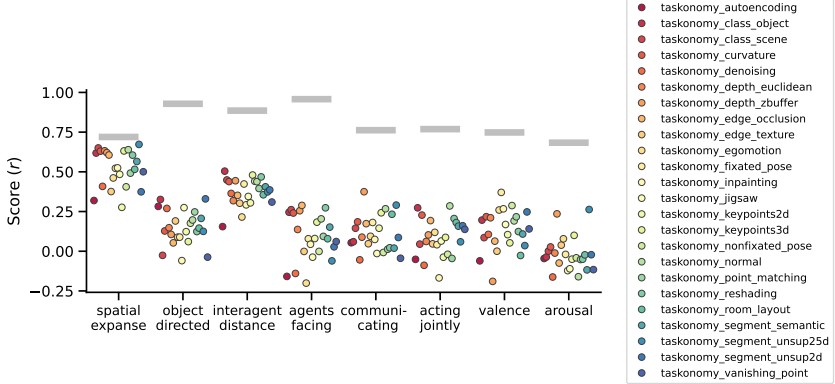

Figure 11: The performance of the taskonomy (Sax et al., 2020; Zamir et al., 2018) models in predicting behavioral responses. Plotting conventions are the same as Figure 2.

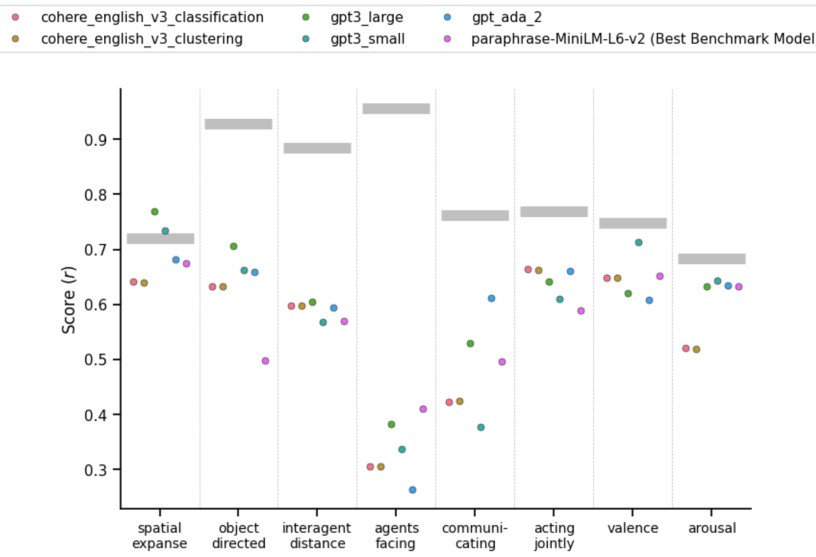

Figure 12: The performance of the GPT-3, GPT Ada-002, and Cohere models compared to the overall best performing model across behavioral ratings. Plotting conventions are the same as Figure 2.

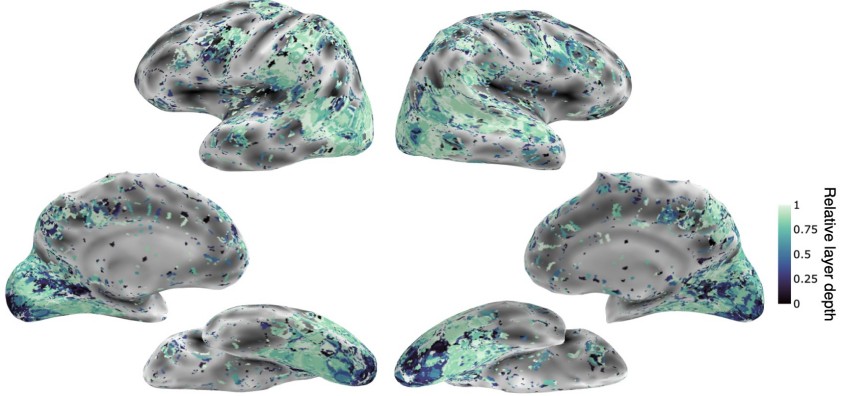

Figure 13: Relative depth of the best performing model layer across all vision models (image and video models) in the whole brain of one representative subject.

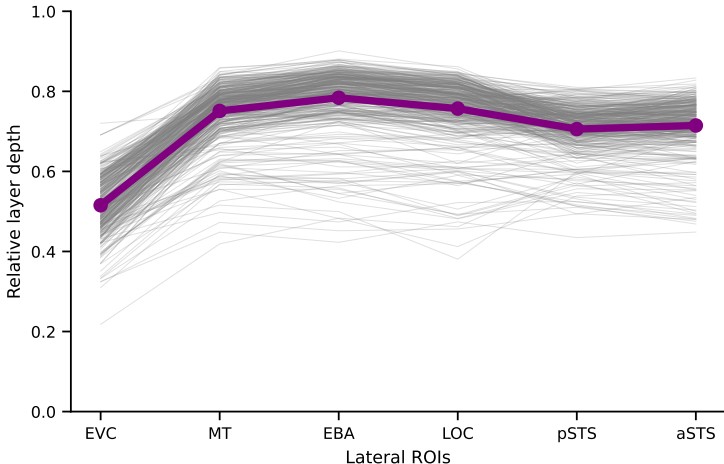

Figure 14: The relative layer depth of the best performing model layer for each image and video model (thin gray lines) and the average best layer depth across models (thick purple line) in each ROI along the lateral visual stream.

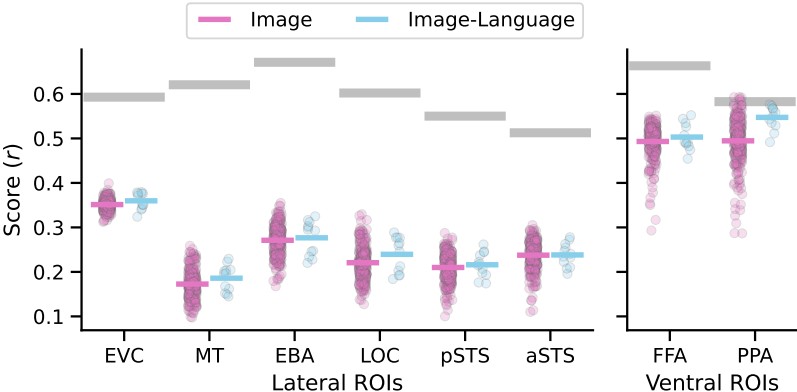

Figure 15: The performance of models in our set trained either with an image-based (image, pink) or multimodal (image-language, blue) objective function in predicting neural responses in ROIs in the lateral and ventral visual streams. Plotting conventions are the same as Figure 4.

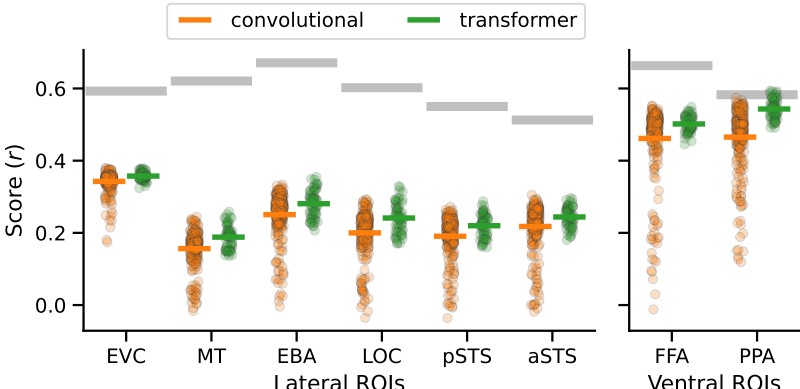

Figure 16: The performance of each image model in our set in predicting neural responses in ROIs in the lateral and ventral visual streams. Models are grouped by convolutional (orange) or transformer (green) architecture. Plotting conventions are the same as Figure 4.

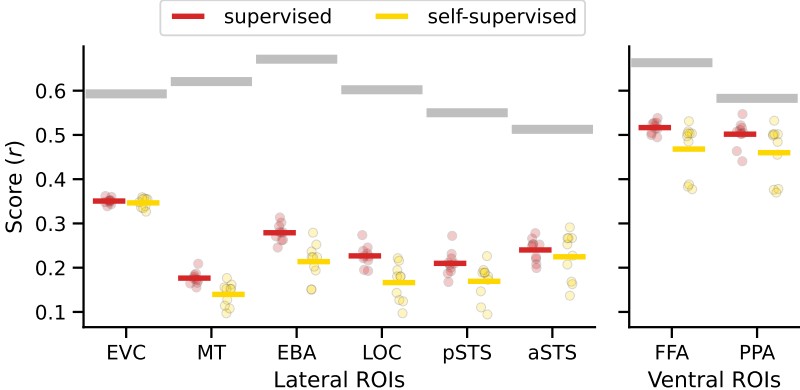

Figure 17: The performance by each image models with a ResNet-50 backbone in predicting neural responses in ROIs in the lateral and ventral visual streams. Models are grouped by supervised (red) or self-supervised (yellow) learning objective. This analysis was restricted to a single architecture class (ResNet-50) to focus specifically on the training objective. Plotting conventions are the same as Figure 4.

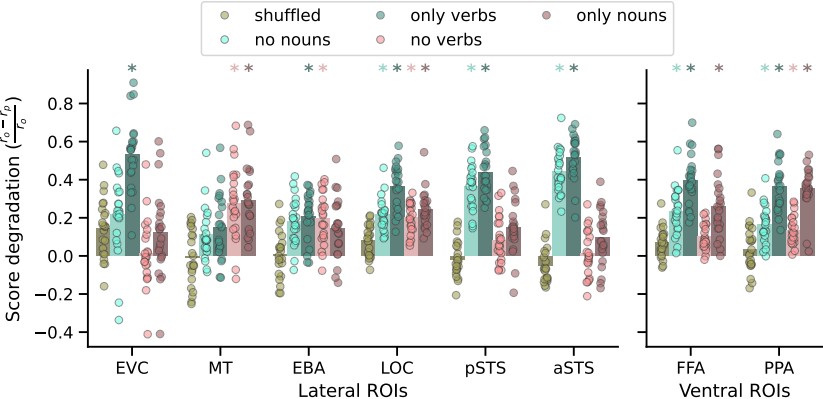

Figure 18: The average performance in ROIs of each language model (dots) in predicting neural responses following selective perturbation of the sentence captions. The bars indicate the mean performance across models for each condition and rating. Asterisks indicate that there is a significant degradation in model-neural alignment following perturbation relative to the unperturbed sentence.

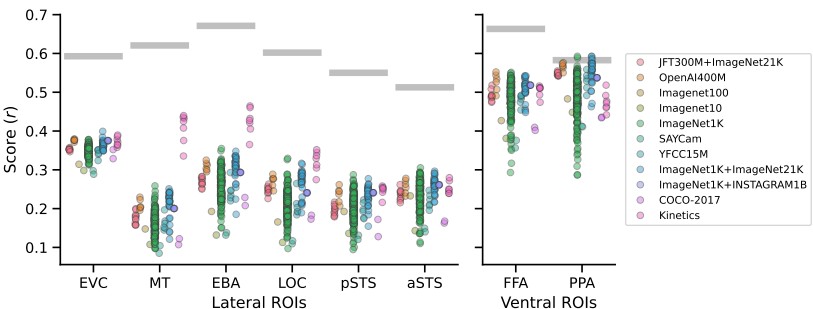

Figure 19: The performance of the vision models (image and video) at predicting neural responses grouped according to their training data. Plotting conventions are the same as Figure 4.

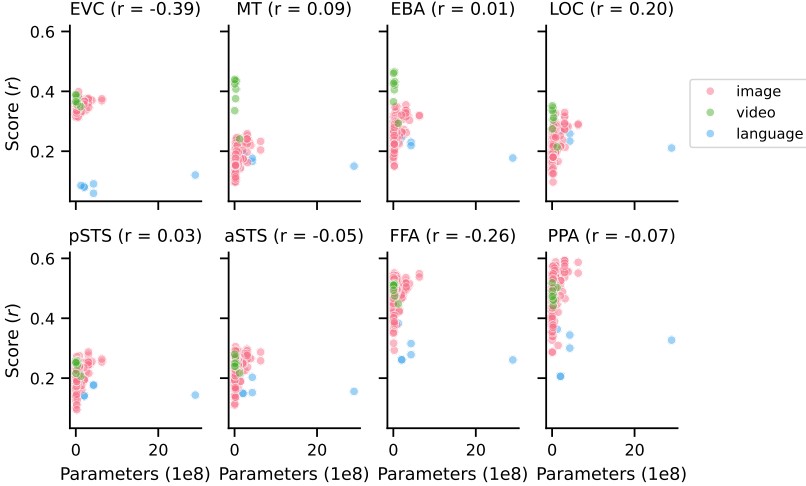

Figure 20: For each ROI response, the model's prediction score is plotted against the number of trainable parameters. The r-value next to the ROI response indicates the Pearson correlation between the score and number of trainable parameters.

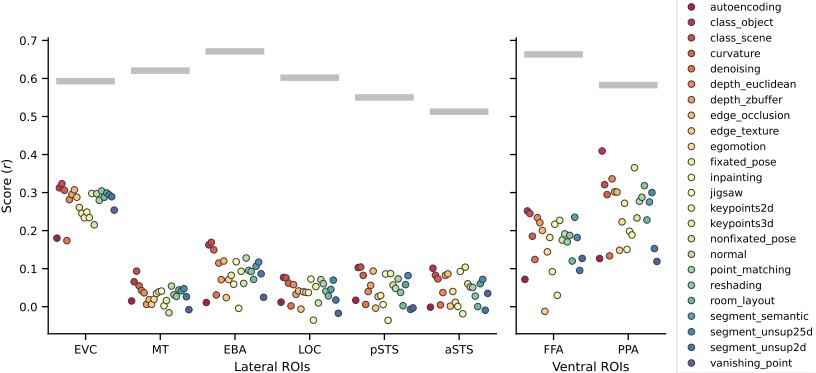

Figure 21: The performance of the taskonomy (Sax et al., 2020; Zamir et al., 2018) models in predicting neural responses in lateral and ventral ROIs. Plotting conventions are the same as Figure 4.

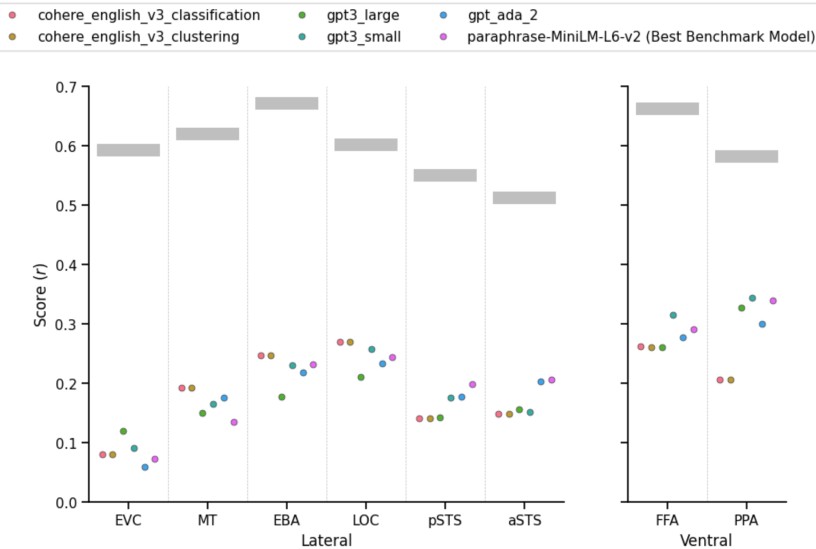

Figure 22: The performance of the GPT-3, GPT Ada-002, and Cohere models compared to the overall best performing model. Plotting conventions are the same as Figure 4.

