# OpenReview forum: "Modeling dynamic social vision highlights gaps between deep learning and humans"
_ICLR.cc/2025/Conference — ICLR 2025 Poster_

### Official Review · Reviewer_tRMU · 2024-10-27

**Soundness:** 3
**Presentation:** 2
**Contribution:** 3
**Rating:** 6
**Confidence:** 2

**Summary:**

Unlike previous neuroscience AI studies that focus on deep learning models' responses to static images, this paper examines models' responses to human social interactions in dynamic videos. The authors create a small benchmark and evaluate an impressive number of models (over 350 image, video, and language models) on this dataset. These comprehensive experimental results could offer valuable insights for researchers in this field.

**Strengths:**

- The relationship between AI models and the human brain is an important area of study.
- This paper presents a highly comprehensive benchmark for examining how AI models—spanning image, video, and language models—respond to social interactions compared to human brain responses in similar scenarios.
- The limitations and discussions offer valuable insights that can inform and guide future research developments.

**Weaknesses:**

- The technical contribution is limited. While I understand the overall effort and workload invested in this paper, it does not address the question of how to develop a human-like AI model.
- Questions regarding the evaluation method: Why is a linear mapping applied between the extracted features and human data, such as fMRI responses? The rationale or assumptions behind this choice of linear mapping are not clear to me. Furthermore, is it appropriate to apply the same linear mapping approach across all models, despite their significant differences? If the evaluation method is not well-justified, the overall efforts should be reassessed carefully.

**Questions:**

- Please include additional details about the experiments.
- It seems to remain unclear whether developing a human-aligned model would enhance existing AI models, why not test some social behavior tasks used in the computer vision community, such as the social interaction tasks in [Ego4D [CVPR2022]](https://arxiv.org/abs/2110.07058) ? (not required to conduct such experiments)

Moreover, given my limited expertise in this specific topic, I recommend that the significance of this paper be assessed by reviewers with specialized knowledge in this area.

---

> ### Author Response · Authors · 2024-11-14
> **Initial review resopnse**
>
> Thank you for your thoughtful and constructive feedback. As noted above we have updated a new version of our manuscript with initial changes. We plan to update a final version with all outlined analyses next week. Please find responses to your individual suggestions below.
>
> Technical contribution: Thank you for raising this issue. We believe the primary advance in this paper is in terms of NeuroAI, and specifically, identifying computational factors needed to match the brain and behavior under dynamic, natural contexts, which have been drastically understudied in this field. In our discussion, we also highlight a couple of insights from our results for better, human-like social AI, namely relational and dynamic representations. We discuss the findings supporting each of these points and possible future directions for better human-aligned social AI.
>
> Linear mapping: We use a linear mapping procedure that is standard in NeuroAI across recording modalities (fMRI, human intracranial, and non-human primate electrophysiology) and input modalities (image, video, language) (e.g., Yamins et al., 2014, Schrimpf et al., 2021, Lahner et al., 2024, Conwell et al., 2024). The intuition for the use of a linear mapping is that if a representation (all activations in a hidden layer of a network) is well suited for a particular task (e.g., judging whether people in a video are communicating), the representation should be predictive of a task up to an affine transformation. Given this reason, the approach is well suited to evaluating models regardless of their differences.
>
> Experimental details: Can you please specify what experimental details you believe are missing from the paper. We note that details for the fMRI and behavior experiments are extensively discussed in the original paper that collected these data (McMahon et al., 2023), and due to space considerations, we included only the elements we believed to be critical for understanding the current work. We are happy to include additional details in the appendix, if you can specify what would be most useful.
>
> Existing computer vision datasets: Ego4D is indeed a great dataset, and it has some overlap with the current dataset, in terms of social action annotations like “who is attending to whom”. However, we believe that human-like social AI must be benchmarked against human social intelligence and neural responses. The strength of our current dataset over computer vision datasets is that (1) the particular annotations were selected based on social dimensions that organize representations in the human mind (McMahon & Isik, 2023; McMahon et al., 2023; Tarhan & Konkle, 2020) and (2) it contains high quality fMRI recordings capturing brain responses to the videos. These advantages allow us to capture human-like social understanding in a way that more directly reflects how people perceive social interactions. We appreciate these comments and, thanks to your suggestion, now expand on this in our paper motivation: we aim to understand dynamic visual responses across the brain, focusing on the lateral visual stream and human annotations of the features of interest for these regions.
>
> References:
> C Conwell, JS Prince, KN Kay, GA Alvarez, T Konkle. A large-scale examination of inductive biases shaping high-level visual representation in brains and machines. Nature Communications 2024, 15 (1), 9383
>
> Benjamin Lahner, Kshitij Dwivedi, Polina Iamshchinina, Monika Graumann, Alex Lascelles, Gemma Roig, Alessandro Thomas Gifford, Bowen Pan, SouYoung Jin, N. Apurva Ratan Murty, Kendrick Kay, Aude Oliva, and Radoslaw Cichy. BOLD Moments: modeling short visual events through a video fMRI dataset and metadata. Nature Communications, 15:6241–6267, March 2023. doi: 10.1038/s41467-024-50310-3
>
> McMahon, E., & Isik, L. (2023). Seeing social interactions. Trends in Cognitive Sciences, 27(12), 1165–1179. https://doi.org/10.1016/j.tics.2023.09.00
>
> McMahon, E., Bonner, M. F., & Isik, L. (2023). Hierarchical organization of social action features along the lateral visual pathway. Current Biology, 33(23), 5035-5047.e8. https://doi.org/10.1016/j.cub.2023.10.015
>
> Tarhan, L., & Konkle, T. (2020). Sociality and interaction envelope organize visual action representations. Nature Communications, 11(1). https://doi.org/10.1038/s41467-020-16846-w
>
> Martin Schrimpf, Jonas Kubilius, Michael J Lee, N Apurva Ratan Murty, Robert Ajemian, and James J DiCarlo. Integrative benchmarking to advance neurally mechanistic models of human intelligence. Neuron, 2020.

---

> > ### Author Response · Authors · 2024-11-22
> > **Updated manuscript**
> >
> > We have posted a revised version of our manuscript incorporating the additional analyses and clarifications based on all reviews. Changes are shown in blue, and major revisions are outlined in our overall rebuttal and responses to your original points are addressed individually above. Thank you again for your helpful feedback. We are eager to hear if you have any remaining feedback.

---

> > > ### Comment · Reviewer_tRMU · 2024-11-25
> > >
> > > I would like to thank the authors for their detailed explanations and responses during the rebuttal.
> > > Based on their significant efforts in both the work itself and the clarification provided, I have decided to revise my rating from borderline reject (BR) to borderline accept (BA).
> > > I remain uncertain whether the contributions presented in this paper are sufficiently impactful for NeuroAI, as it is far from my area of expertise.
> > > This uncertainty is why I am not assigning a higher score.

---

### Official Review · Reviewer_XJaQ · 2024-11-02

**Soundness:** 2
**Presentation:** 2
**Contribution:** 2
**Rating:** 8
**Confidence:** 3

**Summary:**

1. The authors benchmark hundreds of image, video and language models for behavior ratings and neural responses based on human social videos and their captions.
2. The authors compare these models based on their predictions of the behavior rating and neural responses.
3. The authors present several conclusion about the compared models and highlight gaps in their alignment.

**Strengths:**

1. The authors extend a human social video dataset with human annotated text descriptions.
2. The authors benchmark 350 image, video and language models for behavior rating and neural response prediction.
3. The authors highlight gaps in alignment of these models and compare their performance along different axes like architecture, training objective etc.

**Weaknesses:**

1. Just mentioning broad non-exhaustive categories like "self/category supervision, multi-modal and convolution vs transformer" is not a systematic approach to model selection. The authors should first select dimensions of model categorisation they want to compare like supervision type, generative or discriminative, model size, modality etc. and then choose models representing each type. The entire process needs to be detailed to make sure there is no bias in model selection that might affect conclusions downstream. The authors should provide their detailed model selection method.

2. While the authors benchmark several vision models, the vast majority of them are trained for image classification. Models trained with other objectives like object detection (only 2 used as far as I can tell), segmentation (none used as far as i can tell), masked reconstruction (none used as far as i can tell) etc. should also be benchmarked. More generally, vision based models should be categorised based on the different objectives used to train them and compared to provide insights into how different training objectives affect performance/alignment.

3.  While there are several generative language models benchmarked, the vision language models are primarily discriminative. Generative vision models like diffusion based (like stable diffusion), GAN based (like VQ-GAN) and VAE (like VQ-VAE) based models should also be benchmarked for insights into the performance gap between generative and discriminative representations for both language and vision models.

4. The comparisons between the benchmarked models need to be fair in terms of the number of parameters. It looks like the authors have compared the models regardless of the size. The authors should investigate and report whether there exists trends between prediction performance and model size for insights into how model size affects representation quality/alignment.

**Questions:**

see weaknesses

---

> ### Author Response · Authors · 2024-11-14
> **Initial review response**
>
> Thank you for your thoughtful and constructive feedback. We have updated a new version of our manuscript with initial changes. We plan to update a final version with all outlined analyses next week. Please find responses to your individual suggestions below.
>
> Model selection method: Thank you for highlighting this omission. We have clarified this in the model selection section of methods “We used an “opportunistic" modeling approach used in other recent NeuroAI benchmarking research (Conwell et al., 2024 Nature Communications). We selected a large set of publicly available models with a variety of modalities, architectures, training sets and objectives, and evaluated their performance along each of these dimensions.” The primary organizing dimension where we see differences is around image modality, and thus that is the highlight of our main text figures, but the other manipulations are included in the appendix. Based on your suggestion and those from Reviewer jqVp03 we have added pre-training dataset to these analyses. The figures are Appendix Figures 12 and 19. As you note, there are limitations to this approach. In some cases, we have a tight control between all other factors besides the factor of interest (e.g., models with same Resnet backbone and training data but different training objectives, or SLIP family models) but in other cases we do not and acknowledge this limitation in the paper discussion.
>
> Vision model tasks: As with the general oversampling of image models, you are correct that most models are trained on an image classification task. To add breadth to the variety of training model tasks, we are planning to include the taskonomy model family, both because they include a broad range of vision tasks, and also because they all have the same architecture and are trained on the same dataset, so differences across models can be better attributed to the training tasks. Finally, we note that within the self-supervision tasks, there are different forms of contrastive learning (jigsaw, SimCLR, DINO). Overall, we see little difference in neural predictivity between the different self-supervised models or with supervised image classification (Appendix Figure 16).
>
> Generative vision models: It is an interesting idea to compare generative and discriminative vision models. We are wondering if you suggest this due to the theoretical claims about the extent to which human vision is generative (e.g., Yuille and Kersten 2006; Peters et al., 2024). It would be straightforward to add encoder embeddings for generativity trained models like a VAE to our set, but it is unclear these encoders (despite their training) are meaningfully different than discriminatively trained models (see Peters et al., 20224 for a discussion). Alternatively, we could look at embeddings from the generative elements of these networks, such as stable diffusion. We are investigating using DallE embeddings for this purpose, but as this is not an open model we would have to do this by entering our text captions via the API. Please let us know if this would address your concern or if you have another analyses in mind.
>
> Number of tunable parameters: This is an excellent point, and while other recent work (e.g., Linsley et al., 2024 NeurIPS) suggest that larger, performance optimized models may not provide a better match to human neural responses, we plotted our models in terms of human alignment versus number of tunable parameters (Appendix Figures 13 and 20). While we see small correlations, suggesting that on average larger models are slightly better, in all cases, the largest models are not the best models of human data. Thank you for this suggestion!
>
> References:
> C Conwell, JS Prince, KN Kay, GA Alvarez, T Konkle. A large-scale examination of inductive biases shaping high-level visual representation in brains and machines. Nature Communications 2024, 15 (1), 9383
> Drew Linsley, Ivan F. Rodriguez Rodriguez, Thomas Fel, Michael Arcaro, Saloni Sharma, Margaret Livingstone, and Thomas Serre. Performance-optimized deep neural networks are evolving into worse models of inferotemporal visual cortex. Advances in Neural Information Processing Systems, 36:28873–28891, December 2023.
> Peters B, DiCarlo JJ, Gureckis T, Haefner R, Isik L, Tenenbaum J, Konkle T, Naselaris T, Stachenfeld K, Tavares Z, Tsao D, Yildirim I, Kriegeskorte N. How does the primate brain combine generative and discriminative computations in vision? ArXiv [Preprint]. 2024 Jan 11:arXiv:2401.06005v1. PMID: 38259351; PMCID: PMC10802669.

---

> > ### Author Response · Authors · 2024-11-22
> > **Updated manuscript and image model tasks**
> >
> > We have posted a revised version incorporating the planned analyses outlined above. Since our last response, we have added the taskonomy family of models, trained on different objective functions. These are all ResNet-50 architectures with different training objectives ranging from autoencoding to scene recognition to semantic segmentation and surface normals. As shown in Figures 13 and 23, there is variation across task, but all models perform within the range of the main set of our image models and this variation is less than the effect of modality. Finally, we note that within the self-supervision tasks, there are different forms of contrastive learning (jigsaw, SimCLR, DINO). Overall, we see little difference in neural predictivity between the different self-supervised models or with supervised image classification (Appendix Figure 17).
> >
> > Thank you again for your helpful feedback. We are eager to hear if you have any remaining feedback.

---

> > > ### Comment · Reviewer_XJaQ · 2024-11-25
> > >
> > > I thank the authors for their reply.
> > >
> > > Generative vision models: The authors may try to use the internal representations from Stable Diffusion which is open source. This is not a big issue but would be nice to have.
> > >
> > >
> > > Regardless of this result, the authors have addressed almost all my concerns and I have improved my rating.

---

> > > > ### Author Response · Authors · 2024-11-27
> > > >
> > > > Thank you for your positive feedback, and this suggestion. We have added stable-diffusion-v1-4 to our benchmark. For behavior ratings, it is much worse than our best image model (clip vit l14), with the exception of the ratings for: acting jointly, valence, and arousal, where the two models performa similarly. Stable Diffusion is a poor predictor of all neural responses though, and is on the low-end of the performance for all vision models tested. We are working to incorporate this model in our main figures and statistics, which may not be ready by eod, but will be in the final version of our paper.
> > > >
> > > > Thanks again for all of your helpful suggestions!

---

### Official Review · Reviewer_jqVp · 2024-11-03

**Soundness:** 3
**Presentation:** 3
**Contribution:** 3
**Rating:** 8
**Confidence:** 3

**Summary:**

The primary contribution of this work is an analysis of 350 models over an existing dataset of social actions. The labels range from user ratings to fMRI images for brain regions engaged in the watching of videos of social action. The authors find that networks trained over images outperform language/video modalities.

**Strengths:**

1) Action datasets contain both physical and social actions, and are not focused on the exclusive modeling of either action. Exploring models trained over social actions exclusively provides very valuable insights on the difficulty of this domain of action.
2) The evaluation is very broad - 350 models is very impressive. The claims in the discussion are well supported.

**Weaknesses:**

1) The size of the dataset is very small (200 training videos), and the results are definitely impacted by this. Models trained over video datasets in particular must be large due to the variation over the time dimension. The models (and modalities) that perform well might largely be because of the size of the dataset.
2) Audio as a modality is missing, but I would argue is just as valuable as the sequence of images alone across many of the subjects (e.g. valence, arousal). Audio provides less benefit in action datasets focused on physical actions, but might be just as important as the visual modality w.r.t. social actions.
3) There is a lack of discussion around pre-training of the different models. This bears more importance than the model architecture (CNNs vs Transformers) especially due to the small size of the dataset. The video models may or may not be trained over datasets that include social actions (like Kinetics).

**Questions:**

1) I am unfamiliar with the usage of fMRIs in machine learning. But I imagine the variance from individual to individual must cause difficulties in predicting brain responses. Does it not make more sense to condition fMRI prediction on the baseline fMRI readings before the video viewing? Is the text or video input enough?
2) Figure 7 and Figure 8 mention image-language models being evaluated - but to my understanding, all models only take one modality. Are there models that take both images and language?
3) Action datasets deal with label subjectivity - is there any disagreement across annotators concerning the dataset being trained over? I imagine this is particularly a problem in the domain of social action.

---

> ### Author Response · Authors · 2024-11-14
> **Review response**
>
> Thank you for your thoughtful and constructive feedback. As noted above we have updated a new version of our manuscript with initial changes. We plan to update a final version with all outlined analyses next week. Please find responses to your individual suggestions below.
>
> Dataset size: Thank you for highlighting your concerns with the dataset size. We agree that this dataset is too small for training purposes, especially for models with high temporal variation, and we acknowledge this limitation in our discussion. In our current paper this dataset is being used only for model evaluation, not training, which is feasible with this dataset size and comparable to other evaluation datasets (e.g., Conwell et al., 2024 Nature Communications), a point we now mention in our methods section thanks to your vaulable suggestion. This evaluation approach provides a fair basis across models and does not disadvantage any particular model class or modality.
>
> Audio: Thank you for raising this point. We agree that audio is an important component of human social intelligence and can help enhance social action understanding and should be considered in future model evaluations. In our current study, audio was excluded because the human participants in both the behavior and fMRI experiments viewed the videos without sound. As we review in our paper introduction, growing work in cognitive psychology and neuroscience suggest that humans can make rich social judgements from visual information alone (McMahon & Isik 2023 Trends in Cognitive Sciences).
>
> Pretraining: Thank you for highlighting this omission. The datasets that they are trained on vary widely, but these include common computer vision datasets including Kinetics. While the training datasets for each model are included in our all_models_list.csv in the paper supplementary files (to be linked on github upon publication), we agree this point warrants more discussion in the manuscript. To investigate this, we have now included additional appendix plots (Figures 12 and 19) with image and video models separated by training dataset (Imagenet vs. Coco vs YFCC vs. Kinetics vs OpenAI400m). We don’t see meaningful dataset differences for behavior match. For neural match, Kinetics improves performance on most mid-level ROIs, but we note that all kinetics-trained models are video models, so this advantage cannot be disentangled from image versus video performance. We now discuss these results in main text and discussion.
>
> Baseline fMRI responses: In the current approach, we model the voxelwise beta value for each video (compared to rest, output from a generalized linear model). In this way, the baseline normalization you suggest is already taken into account. To further ensure that we are modeling noise-robust responses, we use the test-retest reliability mask provided in the original paper. This restricts our analyses to voxels that show highly similar responses across repeated presentations of the video. (We note as well that subjects see only the videos, not text. The text ratings are collected solely for the purpose of evaluating their match to visually evoked human responses.) Thanks to your suggestions, we have now updated the paper methods to clarify this point.
>
> Multimodal models: Thank you for pointing out the lack of clarity. Some of the models in our set are multi-modal (e.g. CLIP) but you are correct that our neural encoding considers one modality at a time. In Figure 2 and 4, the text encoder of CLIP would be grouped with the language models and the image encoder with the image models. We are happy to inform you that we have now clarified this in the legends of Appendix Figures 7 and 8, as well is in the paper methods: “The vision and language embeddings for multimodal models were considered separately in model evaluation (e.g., CLIP image encoder is grouped with image models and CLIP's text encoder with language models).
>
> Label subjectivity: You are correct that there is variation across annotators, though overall the subject-subject agreement is relatively high (r-values ranging from ~0.7 - 0.9). When learning the linear mapping between humans and models, this variability is ignored by averaging over the responses of all annotators. However, the models are evaluated against the level of human agreement (the gray bars in Figure 2), which takes into account this variability. We consider the level of human agreement to be an approximation of the best that any model could perform in predicting human annotations. We clarified this in the paper methods.
>
> References:
> C Conwell, JS Prince, KN Kay, GA Alvarez, T Konkle. A large-scale examination of inductive biases shaping high-level visual representation in brains and machines. Nature Communications 2024, 15 (1), 9383
>
> McMahon, E., & Isik, L. (2023). Seeing social interactions. Trends in Cognitive Sciences, 27(12), 1165–1179.

---

> > ### Author Response · Authors · 2024-11-22
> > **Updated manuscript**
> >
> > We have posted a revised version of our manuscript incorporating the additional analyses and clarifications based on all reviews. Changes are shown in blue, and major revisions are outlined in our overall rebuttal and responses to your original points are addressed individually above. Thank you again for your helpful feedback. We are eager to hear if you have any remaining feedback.

---

> > ### Comment · Reviewer_jqVp · 2024-11-27
> >
> > Thank you for your work in answering the questions in my review. I will raise my score to accept.

---

### Official Review · Reviewer_RdsD · 2024-11-06

**Soundness:** 3
**Presentation:** 3
**Contribution:** 2
**Rating:** 6
**Confidence:** 3

**Summary:**

In this paper, the author extends a dataset of natural videos describing human action interactions by providing human-annotated sentences for each video and investigates the limitations of over 350+ models to predict human behavioral ratings and neural responses to dynamic social scenes. It concludes that language models predict action and social ratings better than image and video models but perform poorly at predicting neural responses in the lateral stream and provides insights into how well current AI systems replicate human social vision. More importantly, the author highlights the gap in current models' ability to understand dynamic social interactions and suggests potential directions.

**Strengths:**

•	The paper’s approach is innovative in building on the NeuroAI benchmarking with dynamic visual responses rather than using static scene responses, which are commonly evaluated by current image model. It’s the first investigation of benchmarking many models in response to naturalistic videos of human actions.

•	The paper gives a comprehensive model evaluation experiments conducting with this dataset. Spanning from video, language, image models over 350+, including a variety of architectures and objectives.

•	The paper is fully public with all data, code, model accessible. Further, it provides an interesting direction that Human-aligned DNNs may be a promising direction for dynamic social perception, and suggests that developing models that can handle relational and temporal elements essential for social scene understanding.

**Weaknesses:**

•	Limited Coverage or advanced video and language models: Although the dataset has been tested in 350+ models. The majority of them are obsoleted and cannot fully present the overall performance on the state-of-the-art image, video, and language models, like MViT, Co-DETR, DINO, GPT4o, LLAVA, Llama, etc. The most recent model on the paper’s list is up to 2021.

•	The author conducts experiments on a dataset, consisting of 250 3-sec videos, which is relatively small and less representative for training and evaluating deep learning models for making a significant claim on social action recognition. The data limitation might reduce the generalizability of getting conclusions when trading this complex multi-agent interactions task.

•	While the paper make a claim that language models are successful in predicting behavioral response but not in neural responses since providing language captions is insufficient for achieving neural alignment, it encourages the society to make a better connection between neural responses and models with a more well-designed approach. Instead, the conduct of experiments are insufficient in pinpointing precise architectural or training factors that contribute to performance differences by given task.

**Questions:**

Questions are asked in the weakness section.

---

> ### Author Response · Authors · 2024-11-14
> **Initial review response**
>
> Thank you for your thoughtful and constructive feedback. We have updated a new version of our manuscript with initial changes. We plan to update a final version with all outlined analyses next week. Please find responses to your individual suggestions below.
>
> Video/language model coverage: We appreciate your feedback and agree that the limited coverage of advanced video and language models is a limitation of our study (noted in the paper discussion). Because our approach conducts a full layer-wise sweep of each model, we are limited by memory requirements and the need for publicly available models. Within our set, we note that larger and more recent video and language models do not lead to notable improvements (e.g., SBERT outperforms GPT in predicting human social behavior judgements). More recent work in neuroAI benchmarking has also suggested that there is often not an improvement (e.g., Brainscore.org; Linsley et al., NeurIPS 2023). In a new analysis, we also find that models with more tunable parameters are not necessarily more predictive of human responses (Appendix Figures 13 and 20). Finally, we note that some of the image models you mention (e.g., DINO ViT and MViT variants) are already in our existing model set.
>
> Nevertheless, we agree with your suggestion and thus, we would like to include some more modern models in our benchmark, and thus are planning to use the OpenAI API to extract modern language model embeddings of our stimuli. As noted above,  full layerwise sweep is not possible with these models, but we believe this will still be a valuable benchmark to understand to what extent more modern AI models capture visual social judgments.
>
> Dataset size: You are correct that the social action dataset used here is relatively small, which has understandable drawbacks for model training. We acknowledge this point in the paper discussion. Although, notably, the size of the dataset does not limit its potential for model evaluation, the primary purpose of this paper, and it is on par with other datasets used for similar neuro benchmarking purposes (e.g., the shared set of 1000 images in NSD, Conwell et al., 2024 Nature Communications). In this way, the dataset can be thought of as a challenge set that evaluates whether a model has learned human-like generalizable social action representations. Our current findings show that these models have not learned these representations. In fact, in more anterior brain regions, almost all AI models tested perform worse than the simple model reported in the original paper (McMahon et al., 2023). We plan to add the original paper’s encoding model results (32 dimensions compared to the 4732 SRP dimensions from the AI models tested in the current paper) to Appendix Table 2 (final row) to contextualize the results on this dataset.
>
> Pinpointing relevant model factors: Thank you for this comment. We acknowledge in the paper discussion that we are somewhat limited in our ability to make generalizations about what kinds of computations are needed in models for human-like social AI. However, the success of LMs in predicting human social judgments combined with prior models showing that relational-inductive biases allow models to learn social representations (Malik & Isik, 2023 Nature Communications) suggest that future models must learn these relational representations. Perhaps most strikingly, from a NeuroAI perspective, we see a huge advantage of video models (even older models as you point out above) in matching neural responses in mid-level regions, compared to more modern image models. While the AI field has recently expanded its array of available video models, most neuroAI studies of vision still focus on static scene responses. We highlight these points in the “Future Directions” of the current manuscript.
>
> References:
> C Conwell, JS Prince, KN Kay, GA Alvarez, T Konkle. A large-scale examination of inductive biases shaping high-level visual representation in brains and machines. Nature Communications 2024, 15 (1), 9383
>
> Drew Linsley, Ivan F. Rodriguez Rodriguez, Thomas Fel, Michael Arcaro, Saloni Sharma, Margaret Livingstone, and Thomas Serre. Performance-optimized deep neural networks are evolving into worse models of inferotemporal visual cortex. Advances in Neural Information Processing Systems, 36:28873–28891, December 2023.
>
> McMahon, E., Bonner, M. F., & Isik, L. (2023). Hierarchical organization of social action features along the lateral visual pathway. Current Biology, 33(23), 5035-5047.e8.
>
> Manasi Malik and Leyla Isik. Relational visual representations underlie human social interaction recognition. Nature Communications, 14(1):7317, November 2023.

---

> > ### Author Response · Authors · 2024-11-22
> > **Updated paper and gpt results**
> >
> > We have posted a revised version of your manuscript with the changes outlined in blue. We now include more modern language models that we extract via the OpenAI and Cohere APIs to extract embeddings from GPT3 small, GPT3 large, GPT-Ada, and Cohere English-v3 variants (though we note that due to the closed nature of these models, a full layer sweep is not feasible). Interestingly, there are no major differences between these models and the best language model from our original set in predicting behavior or neural responses (Tables 1 and 2, Figures 14, 24).
> >
> > Thank you again for your helpful feedback. We are eager to hear if you have any remaining suggestions.

---

> ### Comment · Reviewer_RdsD · 2024-11-27
>
> I want to thank the authors for their hard work during the rebuttal period. I will raise my rating based on your response that a huge advantage of video models (even older models as you point out above) is in matching neural responses in mid-level regions, compared to more modern image models from a NeuroAI perspective and also the extra experiments working on OpenAIs' LLM. However, I still remain my uncertainty on the selection of advanced VLMs and LLMs and also remain unclear for whether conducting experiments within limit number of data will lead to a generalizable result on this topic. Regarding to your analysis mentioning that you find that models with more tunable parameters are not necessarily more predictive of human responses, I still think the main reason might cause by the training process and also the size of the dataset by those given task.
>
> Again, thanks for your contribution on human behavioral and neural responses to videos of social actions from a NeuroAI perspective.

---

> > ### Author Response · Authors · 2024-11-27
> >
> > Thank you for your positive feedback.
> >
> > You raise a good point about training dataset size. Based on this concern and the suggestion of jqVp, we analyzed performance of the vision models trained on different datasets (Imagenet vs. Coco vs YFCC vs. Kinetics vs OpenAI400m). The results are included in appendix plots (Figures 12 and 19). We don’t see meaningful dataset differences for behavior match based on training dataset. For neural match, Kinetics improves performance on most mid-level ROIs, but we note that all kinetics-trained models are video models, so this advantage cannot be disentangled from image versus video performance. We now discuss these results in main text and discussion.
> >
> > Thank you again for your helpful suggestions. We believe the manuscript is much stronger thanks to feedback from you and the other reviewers.

---

### Author Response · Authors · 2024-11-14
**General review response**

We are grateful to the reviewers for their thoughtful and constructive feedback on our manuscript. We have updated our manuscript with additional analyses and edits we made based on your feedback.

We are still running some analyses to extract embeddings for additional models, including models trained on different image tasks and larger language models, and plan to upload another version with these updates next week. In the meantime, we are happy to hear your feedback on these revisions and if you have additional suggestions for how we can further improve the paper together. Thank you!

---

> ### Author Response · Authors · 2024-11-22
> **Updated paper and overall review response**
>
> We thank the reviewers again for their positive and constructive reviews. We have now made several major changes based on their feedback that we outline below (blue text in revised pdf). We believe the manuscript is significantly improved thanks to this feedback and we hope the reviewers agree.
>
> 1. Clarified and expanded opportunistic modeling approach [Reviewers jqVp, XJaQ]: We have clarified the overall goals of our opportunistic model approach in the manuscript: “We used an “opportunistic" modeling approach used in other recent NeuroAI benchmarking research (Conwell et al.). We selected a large set of publicly available models with a variety of modalities, architectures, training sets and objectives, and evaluated their performance along each of these dimensions.” The primary organizing dimension where we see differences is around image modality, and thus that is the highlight of our main text figures, but the other manipulations are included in the appendix.
>
> 1b. We have also expanded our analyses to include image pre-training dataset (Figure 11 and 21) and number of tunable parameters (Figure 12 and 22). Neither of these factors make a meaningful difference in neural predictivity relative to the overall effect of modality.
>
> 2. Added new models to the benchmark
> 2a. Modern models [Reviewer RdsD] We first clarify that our dataset includes several more modern image models (e.g., BeIT, DINO variants). As we note in the discussion, the full layer sweep of our approach restricts our main analyses to publicly available models without large memory requirements, so we now use the OpenAI and Cohere APIs to extract embeddings from GPT3 (small and large), GPT-Ada, and Cohere English-v3 variants (though we note that due to the closed nature of these models, a full layer sweep is not feasible). We find that these models performs similarly to the best language model in our current set. While there are some slight advantages for some behavior ratings, none of these are in the top 30 models on any neural benchmark (Table 1 and 2, Appendix Figures 14 and 24).
>
> 2b. Models with diverse objective functions (Reviewer XJaQ): To test image models with more diverse objective function, we added the set of Taskonomy models to our set of models (Zamir et al., 2018). These are all ResNet-50 architectures with different training objectives ranging from autoencoding to semantic segmentation and surface normals. As shown in Figures 13 and 22, there is variation across task, but all models perform within the range of the main set of our image models and this variation is less than the effect of modality. Finally, we note that within the self-supervision tasks, there are different forms of contrastive learning (jigsaw, SimCLR, DINO). Overall, we see little difference in neural predictivity between the different self-supervised models or with supervised image classification (Appendix Figure 18).
>
> 3. Clarified details of fMRI measurements and model-brain mapping
> 3a. fMRI responses [Reviewer jqVp, tRMU]: We have added details to the paper methods to clarify the fMRI responses we are modeling (beta responses to each video, relative to baseline, as measured by a generalized linear model).
>
> 3b. Dataset size and model evaluation [Reviewers RdsD, jqVp]: The reviewers are correct to point out that the social action dataset used here is relatively small. The dataset size is a limitation for model training, which we acknowledge in the paper discussion. We note though, that size of the dataset does not limit its potential for model evaluation, which is the primary purpose of this paper. The dataset size is on par with other datasets used for similar neuroAI benchmarking (e.g., Conwell et al., 2024 Nature Communications). In this way, the dataset can be thought of as a challenge set that evaluates whether a model has learned human-like generalizable social action representations. Our current findings show that these models have not learned these representations. In fact, in more anterior brain regions, almost all AI models tested perform worse than the simple model reported in the original paper (McMahon et al., 2023), the performance of which we have added to Appendix Table 2 (final row) to contextualize the results on this dataset.
>
> 3c. Linear mapping [Reviewer tRMU]:  We use a linear mapping procedure that is standard in NeuroAI across recording modalities (fMRI, human intracranial, and non-human primate electrophysiology) and input modalities (image, video, language) (e.g., Yamins et al., 2014, Schrimpf et al., 2021, Lahner et al., 2024, Conwell et al., 2024). The motivation is that if a representation (all activations in a hidden layer of a network) is well suited for a particular task (e.g., judging whether people in a video are communicating), the representation should be predictive of a task up to an affine transformation. Related to the above point, this approach is also well suited for a dataset of this size.

---

### Meta-Review · Area_Chair_Ar9E · 2024-12-20

**Metareview:**

This paper investigates the ability of deep learning models to predict human behavioral ratings and neural responses, proposing a dataset of human action-interactions to build on existing simpler image-based works. The dataset includes human annotated captions, allowing for use of language models as well. Over a large number of models, spanning image, video, and language models, a number of findings are reached, including the inability of vision models to predict human action and social interactions and neuronal responses along the ventral visual stream, the improvements in predicting mid-level lateral stream activity by video models, and improved ability of language models to predict action and social ratings. Overall, the proposed benchmark and large-scale experimentation seeks to improve the understanding of how current models do and do not match representations in the brain.

  The reviewers all appreciated the idea of benchmarking NeuroAI tasks using more complex, dynamic datasets and responses rather than static scenes, as well as the comprehensive nature of the code. However, a number of shared concerns were raised including limited coverage of recent models (after 2021) as well as lack of a coherent framing/categorization for their selection, limitations on the amount/richness of data in the dataset, and lack of other modalities such as audio (which are important in human processing of real-world data). As part of the rebuttal, the authors incorporated a number of additional models including Taskonomy models and GPT models by OpenAI/Cohere (via APIs), and diffusion models. Additionally, text including justification of model selection and other details were added. Overall, reviewers were satisfied that many of their concerns were addressed.

  Considering all of the materials, I recommend acceptance of this paper. While there are still some limitations, the current version would be a benefit to the community in terms of providing a richer benchmark for this task and analysis across a range of models. Namely, the addition of more modern GPT and diffusion models significantly added to the impact of the paper. I highly encourage the authors to include the model selection justification and additional results in the camera-ready version.

**Additional Comments On Reviewer Discussion:**

Reviewers had significant back-and-forth with the authors and noted that most of their concerns were addressed. Reviewer RdsD raised their score from 5 to 6.

---

### Decision · Program_Chairs · 2025-01-22

Accept (Poster)